# TRIM7 Restricts Coxsackievirus and Norovirus Infection by Detecting the C-Terminal Glutamine Generated by 3C Protease Processing

**DOI:** 10.3390/v14081610

**Published:** 2022-07-23

**Authors:** Jakub Luptak, Donna L. Mallery, Aminu S. Jahun, Anna Albecka, Dean Clift, Osaid Ather, Greg Slodkowicz, Ian Goodfellow, Leo C. James

**Affiliations:** 1MRC Laboratory of Molecular Biology, Francis Crick Avenue, Cambridge CB2 0QH, UK; jluptak@mrc-lmb.cam.ac.uk (J.L.); dlm@mrc-lmb.cam.ac.uk (D.L.M.); aam@mrc-lmb.cam.ac.uk (A.A.); dclift@mrc-lmb.cam.ac.uk (D.C.); oa260@cam.ac.uk (O.A.); 2Division of Virology, Department of Pathology, University of Cambridge, Addenbrooke’s Hospital, Cambridge CB2 0QQ, UK; asj40@cam.ac.uk (A.S.J.); ig299@cam.ac.uk (I.G.); 3The Francis Crick Institute, London NW1 1AT, UK; grzegorz.slodkowicz@crick.ac.uk

**Keywords:** TRIM7, coxsackievirus, norovirus, 3C protease, 3Cpro, Mpro, SARS-CoV-2, restriction, degradation, ISG

## Abstract

TRIM7 catalyzes the ubiquitination of multiple substrates with unrelated biological functions. This cross-reactivity is at odds with the specificity usually displayed by enzymes, including ubiquitin ligases. Here we show that TRIM7′s extreme substrate promiscuity is due to a highly unusual binding mechanism, in which the PRYSPRY domain captures any ligand with a C-terminal helix that terminates in a hydrophobic residue followed by a glutamine. Many of the non-structural proteins found in RNA viruses contain C-terminal glutamines as a result of polyprotein cleavage by 3C protease. This viral processing strategy generates novel substrates for TRIM7 and explains its ability to inhibit Coxsackie virus and norovirus replication. In addition to viral proteins, cellular proteins such as glycogenin have evolved C-termini that make them a TRIM7 substrate. The ‘helix-ΦQ’ degron motif recognized by TRIM7 is reminiscent of the N-end degron system and is found in ~1% of cellular proteins. These features, together with TRIM7′s restricted tissue expression and lack of immune regulation, suggest that viral restriction may not be its physiological function.

## 1. Introduction

TRIM proteins comprise the largest family of E3 ligases in mammals. They are characterized by a tripartite motif of RING, B-Box and coiled-coil and may encode additional domains, of which the PRYSPRY domain is the most common. Many TRIM proteins have proposed roles in infection and immunity (for recent review, see [1]) where they typically inhibit viral replication, although some are reported to promote replication [2,3]. A commonly reported mechanism is one in which the TRIM binds a viral protein and causes it to be degraded via ubiquitination. Targeting is usually mediated by the C-terminal PRYSPRY domain, whilst ubiquitination is catalyzed by the N-terminal RING domain, though TRIMs have been reported to degrade their target even when they lack a RING [4]. Typical targets for degradation include the viral capsid or nucleocapsid: TRIM5 [5], TRIM11 [6], and TRIM34 [7] are reported to target the capsid of HIV and TRIM14 [4], TRIM22 [8] and TRIM41 [9] the nucleocapsid of Influenza.

Antiviral TRIMs have also been shown to stimulate immune signaling, providing a second mechanism of viral inhibition. This can be indirect, by promoting the activity of pattern-recognition receptors (PRRs). For instance, TRIM65 is reported to K63-polyubiquitinate MDA5 and promotes its signaling [10]. RIPLET, technically not a TRIM but containing RING, coiled-coil and PRYSPRY domains, modifies RIG-I with K63-chains and stimulates its activity [11,12], while TRIM56 is implicated in DNA sensing by promoting the activity of cGAS [13]. Other TRIMs promote signaling directly as primary PRRs. TRIM5 [14] and TRIM21 [15] both stimulate innate immune signaling upon detection of incoming virions. This activity occurs in parallel with their direct-acting restriction mechanisms, in which targeted viral capsids are degraded. The restriction and signaling activities of both TRIM5 and TRIM21 are inextricably linked—synthesis of K63-polyubiquitination leads to proteasomal recruitment, followed by concomitant liberation of free K63-chains and protein degradation [16,17].

Exactly how many members of the TRIM family are involved in antiviral immunity, what proportion directly restrict or promote immune signaling and which are capable of both is unknown. In a transient over-expression screen of 36 human and 19 mouse TRIMs, over one-third were found to possess antiretroviral activity [18]. In a follow-up study, 16 out of 43 over-expressed human TRIMs were shown to induce NF-κB or AP-1 signaling [19]. Tellingly, there was a close correspondence between restriction and signaling; of 17 TRIMs that restricted MLV, 12 were also immune stimulatory. A more recent over-expression screen of 118 RING E3 ligases (including over 50 TRIMs) identified TRIM7 as a potent inhibitor of picornavirus Coxsackievirus B3 (CVB3) [20]. Restriction required PRYSPRY binding to viral protein 2BC, followed by its RING-dependent ubiquitination and degradation. TRIM7 was also previously identified in a genome-wide overexpression screen as an inhibitor of the picorna-like virus calicivirus murine norovirus (MNV), an activity similarly found to be PRYSPRY-dependent [21]. However, these reports of antiviral activity are at odds with other TRIM7 studies. TRIM7 has been proposed as a pro-viral factor in Zika virus infection, where it was shown to K63-polyubiquitinate the viral envelope protein [3]. TRIM7 has also been reported as an agonist of TLR4-mediated signaling in macrophages [22], a tumour suppressor through degradation of Src [23], an oncogene through the K63-ubiquitination and stabilization of AP-1 co-activator RACO-1 [24] and an interactor and regulator of glycogenin—the latter activity giving rise to TRIM7s original designation as ‘glycogenin-interacting protein’ (GNIP) [25]. There is little that connects these disparate cellular roles.

In this study, we sought to determine how TRIM7 is capable of interacting with such diverse substrates and the mechanism by which it targets viral proteins for degradation. We show that the PRYSPRY domain of TRIM7 possesses a unique binding site that promiscuously interacts with proteins containing a C-terminus ending in a helix-ΦQ motif. The reason why TRIM7 has been reported to bind glycogenin-1 (GYG1), RACO-1 and CVB3 2BC is because they all possess this structural signature. We also show that it is this promiscuous helix-ΦQ motif that allows TRIM7 to restrict infection by both norovirus and Coxsackievirus. These viruses possess a viral 3C-protease that cleaves their polyproteins to generate viral proteins ending in glutamine. A TRIM7 point mutant that no longer binds this glutamine motif loses all viral restriction. 3C-like proteases are common to positive-strand RNA viruses from the Picornaviridae, Caliciviridae and Coronaviridae families, and all generate viral proteins with C-terminal glutamines that are potential substrates for TRIM7. However, we show that despite possessing multiple proteins ending in glutamine, SARS-CoV-2 is not restricted by TRIM7. Moreover, the tissue expression of TRIM7 and its gene regulation is not consistent with an antiviral TRIM, such as TRIM5 or TRIM21, suggesting that its described function as a restriction factor may be an artefact of overexpression.

## 2. Material and Methods

### 2.1. Proteins

Trim7 PRYSPRY constructs and all GYG constructs with an N terminal hexahistidine tag were expressed in *E. coli* (C41) and purified using Nickel affinity chromatography and Size Exclusion Chromatography (SEC). Briefly, cells were grown in 2XTY (supplemented with 0.5% glucose, 2 mM MgSO_4_ and appropriate antibiotics) at 37 °C for 2–3 h (OD600 around 0.6–1), after which they were induced with 1 mM IPTG and incubated at 18 °C overnight. Cells were pelleted with a Sorvall SLC-6000 compatible centrifuge at 4500× *g* for 25 min and the pellet frozen until processed. The pellet was resuspended in lysis buffer (50 mM Tris pH 8, 1 M NaCl, 10% *v*/*v* BugBuster (Merck, Gillingham, UK), 10 mM imidazole, 2 mM DTT and 1 × complete protease inhibitors (Roche, Basel, Switzerland) and sonicated for 15 min total time (10 s on/20 s off) at 70% amplitude. The soluble fraction was recovered by centrifugation at 40,000× *g* in a JLA25.50 rotor and put through a gravity flow column with 5 mL of NiNTA Agarose (Qiagen). The bound fraction was washed in Buffer B (300 mM NaCl, 50 mM Tris pH 8, 10 mM imidazole and 1 mM DTT) and eluted with Buffer E (300 mM NaCl, 50 mM Tris pH 8, 400 mM imidazole and 1 mM DTT). Fractions containing the protein were pooled, filtered and separated by SEC using HiLoad 26/600 Superdex 75 pg column (Cytiva, Marlborough, MA, USA) in 150 mM NaCl, 50 mM Tris pH 8 and 1 mM DTT. Full-length GYG proteins were separated using an equivalent 200 pg column. The appropriate fractions were pooled and concentrated to 10–15 mg/mL.

MBP-Trim7-CCPS was expressed and purified as described above but with the following adjustments; after elution from NiNTA agarose, the fractions were pooled and loaded onto an MBP-Trap (5 mL) column (300 mM NaCl, 50 mM Tris pH 8, 1 mM DTT) washed with the same buffer and then eluted with buffer supplemented with 10 mM Maltose. The peak fractions were then separated by SEC using HiLoad 26/600 Superdex 200 pg column in 300 mM NaCl, 50 mM Tris pH 8, 10% *w*/*v* glycerol and 1 mM DTT. The appropriate fractions were pooled and concentrated to 20 mg/mL.

TRIM7-RING and TRIM7-RING-Box were expressed in *E. coli* C41 cells as GST–TEV fusion protein. Cleared cell lysates were prepared by sonication in 50 mM Tris at pH 8 (pH 9 for RING-Box), 150 mM NaCl, 2 mM DTT with the addition of 20% (*v*/*v*) BugBuster and complete protease inhibitors, followed by centrifugation 16,000× *g* for 30 min. Lysates were loaded onto GST beads and washed with lysis buffer, then cleaved with TEV protease overnight at 4 °C. Cleaved proteins were concentrated and separated using a HiLoad 26/600 Superdex 75 size exclusion column. The peak fractions were pooled, concentrated and frozen in aliquots at −80 °C. Ube1, Ube2N, Ube2V2, Ube2W were produced as previously described [26].

### 2.2. Constructs and Cloning

TRIM7-PRYSPRY constructs (both mouse and human) were ordered as synthetic genes encoding the residues 342–511 of TRIM7 and cloned into a double-digested pOPTH (introduces uncleavable MAHHHHHHM sequence at the N terminus) expression vector (KpnI, NdeI). MBP-hTRIM7-CC-PS was cloned by amplifying the hTrim7-PRYSPRY sequence and Gibson assembly with a synthetic gene fragment containing the CC sequence (residues 165–341) into a double-digested pOPTM (introduces hisMBP-TEVsite at the N terminus) expression vector (KpnI, NdeI). TRIM7-RING (residues 1–123) and TRIM7-RING-BOX (residues 1–168) constructs were cloned by PCR amplification from codon optimized (*E. coli*) synthetic DNA and Gibson assembly into double-digested (NdeI, HindIII) pOPTG expression vector (introduces GST-TEVsite at the N terminus).

The pEXN-2XHA plasmid backbone was used in comparison of signaling of different TRIM proteins. pEXN-2XHA-TRIM7 was generated by inserting a synthetic gene encoding full-length codon optimized TRIM7 sequence (GeneArt) into a double-digested pEXN-2XHA vector (EcoRI, XhoI). pEXN-2XHA-TRIM5 was used as previously described [27]. pEXN-2XHA-TRIM21 was cloned from constructs published previously [27]—sequence encoding full-length human TRIM21 was PCR amplified and inserted into double-digested pEXN-2XHA vector (EcoRI, XhoI).

pSMPP-2xHA-TRIM7 constructs were cloned from the pEXN-2XHA-TRIM7 plasmid. Relevant fragments (FL, ΔPRYSPRY(1–341), ΔRING(128–511)) were PCR amplified and cloned into BamHI linearized pSMPPv1 using Gibson assembly. Untagged pSMPP-TRIM7 constructs (FL, ΔPRYSPRY(1–341), ΔRING(128–511)) were ordered as synthetic gene fragments (GeneArt) designed for Gibson assembly into double-digested (NotI, BamHI) pSMPP vector. N terminally mCherry tagged Trim7 constructs were generated from pSMPP-2XHA-Trim7 by PCR amplification and Gibson assembly into double-digested (XhoI, KpnI) pmCherry-C1 plasmid vector.

The pmEGFP-GYG1 plasmid was generated by restriction cloning of a synthesized gene fragment encoding the hsGYG1-GN1 sequence into the pmEGFP-C1 plasmid vector (BsrG1, EcoR1). For bacterial expression of hGYG1-GN1, the plasmid pOPTH-hGYG1-GN1 was generated by Q5 PCR amplification of the sequence from pmEGFP-GYG1 and Gibson assembled into a double-digested pOPTH (introduces uncleavable MAHHHHHH sequence at the N terminus) plasmid vector (KpnI, NdeI). The rest of the GYG1 protein constructs were ordered as synthetic genes and cloned into double-digested pOPTH (KpnI, HindIII). Plasmids encoding rbGYG1 and mGYG1 were ordered as synthetic genes encoding the full-length proteins and cloned into the pOPTH plasmid vector using restriction cloning. Truncated constructs rbGYG1VR-CIV, rbGYG1CIV, rbGYG11-246, rbGYG11-299 were generated from pOPTH-rbGYG1 by PCR amplification and restriction cloning into the same pOPTH vector.

All site-directed mutagenesis was performed with a Q5 PCR kit apart from mutagenesis of pSMPP-TRIM7-R385A, which was cloned by Gibson mutagenesis following manufacturer’s instructions (NEB).

### 2.3. Cell Culture

HEK293T, BV-2 and HeLa-CD300lf cells were grown in Dulbecco’s modified Eagle’s medium (DMEM) supplemented with 10% FBS, 2 mM L-glutamine, 100 U/mL penicillin and 100 mg/mL streptomycin (GIBCO) at 37 °C with 5% CO_2_. U2OS cells were grown in McCoy-5a medium supplemented with 10% FBS, 1XGlutaMAX, 100 U/mL penicillin and 100 mg/mL streptomycin.

### 2.4. NF-κB and AP-1 Signalling Experiments

HEK293T cells were seeded at 1 × 10^5^ cells per well in 24 well plates 24 h before transfection. Each well was transfected with the indicated amounts of pEXN-2XHA-TRIM5/7/21 plasmids, together with 10 ng pGL4.32[luc2P/NF-κB-RE/Hygro] (NF-κB response element-dependent firefly luciferase, Cat. no 9PIE849) or pGL4.44 [luc2P/AP-1-RE/Hygro] (AP-1 response element-dependent firefly luciferase, Cat. no 9PIE411) and 5 ng pRL-TK (Cat. no E2241) using the Dual Luciferase Reporter Assay (Promega, Madison, WI, USA). Twenty-four hours post-transfection, cells were lysed in Passive Lysis Buffer, and sequential firefly and Renilla luminescence measured (BMG PHERAstar plate reader), according to manufacturer’s instructions (Promega). Firefly luciferase luminescence was normalized to Renilla luciferase luminescence, and these values normalized to those of empty vector-transfected cells.

### 2.5. qPCR Signalling Experiments

HEK293T cells were seeded at 2.5 × 10^5^ cells per well in a 6-well plate. The next day the cells were transfected with 250 ng of either pEXN-2XHA-TRIM7 or empty pEXN-HA vector, and the cells harvested after 8 h. RNA was extracted using the RNeasy mini kit (Qiagen, Hilden, Germany) and cDNA produced from 1 µg RNA using oligo dT and Superscript RT. Gene expression was monitored by TaqMan Gene Expression Assays (Applied Biosystems, Waltham, MA, USA) on a StepOnePlus Real-Time PCR System (Life Technologies, Carlsbad, CA, USA). Gene expression assays were from Applied Biosystems: ACTB (Hs01060665_g1), CXCL10 (Hs01124251_g1). Relative expression was quantified using the 2−ΔΔCt method.

### 2.6. Stable Cell Line Generation

Stable cell lines were generated by stable lentiviral transduction of recipient cell lines. To generate the relevant lentiviruses, the untagged pSMPP-TRIM7 constructs were co-transfected into HEK293T cells with pMDG2 (a gift from D. Trono [EPFL; Addgene, plasmid 12259]) and pCRV-GagPol (a gift from S. Neil, Kings College London) plasmids using FuGENE6 (Promega) reagent and incubated for 3 days. Supernatants were harvested by filtration and used for transduction of HeLa cells expressing the mouse CD300lf [28] as previously described [20]. Briefly, 2.5 × 10^5^ cells were infected with the appropriate lentivirus in a 6-well plate in DMEM supplemented with 3% FBS, 20 mM HEPES and 4 μg/mL Polybrene, and spinoculated at 1000× *g* and 25 °C for 45 min. The cells were then incubated at 37 °C for 48 h. Antibiotic selection was carried out with 2 μg puromycin until all untransduced control cells were dead. For SARS-CoV-2 infection experiments, HeLa cells expressing TRIM7 constructs and mouse CD300lf were further transduced with lentiviruses encoding the ACE2 receptor [29] and selected with 10 µg/mL of Blasticidin.

### 2.7. CVB3 Experiments

For production of Coxsackievirus, plasmid encoding for strain pH3 with an eGFP (eGFP-CVB3) (kind gift from Professor Charles Rice) was transfected into HEK293T cells using PEI. After 48 h, stock was harvested by 3 cycles of freeze-thawing and centrifugation to remove the debris. Collected stock (P0) was used to generate working stock (P1) by infecting fresh plate for HEK293T cells and harvesting as above. Virus titer was assessed by TCID50 using HeLa cells. For infection experiments, Hela-CD300lf stably expressing different TRIM7 constructs were seeded onto 24-well plates 24 h prior to infection. Cells were infected at MOI of 0.2, and infection levels determined by measuring eGFP fluorescence after 12 h on the incuCyte.

### 2.8. MNV1 Experiments

HeLa-CD300lf-TRIM7 cells suspended in DMEM supplemented with 10% FBS were incubated with the virus on an end-to-end rotor at 37 °C for 1 h. The cells were then washed twice in DMEM to remove the unbound virus and then incubated at 37 °C for the duration of the infection. The plate (both cells and supernatants) was frozen at −80 °C for 24 h post-infection, and TCID50 assays were subsequently carried out in BV2 cells as previously described [30].

### 2.9. SARS-CoV-2 Experiments

HeLa cells expressing versions of TRIM7, CD300lf and ACE2 were seeded into 96-well plates at a density of 1.5 × 10^4^ day prior to infection. For infections, we used isolate of omicron variant BA.1 (EPI_ISL_7886688), which was a gift from Ravi Gupta (University of Cambridge). Media containing 2% FBS and virus at MOI = 0.2 PFU/cell was added to wells and incubated for 18 h. Cells were freeze-thawed and then lysed with buffer containing 0.25% Triton X-100, 50 mM KCl, 100 mM Tris–HCl pH 7.4, glycerol 40% and RNAsecure (1/100) for 5 min. Infectious virus was then inactivated at 95 °C for 5 min, and samples processed for RT-qPCR as described previously [31]. We used Luna Universal Probe One-Step kit following manufacturer’s recommendations. Primer/probe for genomic viral RNA were CDC-N1. Primer probes for 18S control were described previously [32]. SARS-CoV-2_N_Positive control RNA was used as standard for the viral genomic N reactions. For 18S standard, DNA was synthesized and kindly gifted by Jordan Clarks and James Stewart (University of Liver-pool). Final concentrations of 500 nM for each primer and 125 nM for the probe were used. RT–qPCRs were run on ABI StepOnePlus PCR System with the following program: 55 °C for 10 min, 95 °C for 1 min and then 40 cycles of 95 °C denaturation for 10 s and 60 °C extension for 30 s. RNA copy numbers were obtained from standards, and then genomic copies of N normalized to 1010 copies of 18S.

### 2.10. Co-Localisation and Degradation Experiment

U2OS cells were seeded into 8-well μ-slide (ibidi, polymer) for co-localization or into standard tissue culture 24-well plates for bulk fluorescence quantification and protein quantification. For the former, 2–3 × 10^4^ cells were seeded, and 5 × 10^5^ cells were seeded in the latter. The next day, cells were co-transfected by a 1:1 mixture of the relevant plasmids using FuGENE6 and placed in the incuCyte (Sartorius, Göttingen, Germany) for live imaging. After 18–20 h of transfections, the cells were harvested, and the cell pellets processed as described below. To quantify the degradation of the EGFP-GYG1 using bulk fluorescence by incuCyte, the total integrated intensity in each condition was divided by the green fluorescence area to account for different transfection efficiencies. Co-localization experiment was imaged using live-cell imagining with the LUMASCOPE LS720 (Etaluma, Carlsbad, CA, USA). Cells expressing low levels of each plasmid were selected for analysis to avoid overexpression artefacts and aggregated proteins. Images were processed in ImageJ, and data plotted in PRISM (GraphPad Software, San Diego, CA, USA).

### 2.11. Protein Quantification Using Capillary Based Western Blot (Simple Western)

Cells were lysed in 40 µL RIPA buffer supplemented with 1 × complete protease inhibitors (Roche, Basel, Switzerland), and the soluble protein recovered by centrifugation (40 min at 21 k × g); 30 µL of the lysate was diluted with 90 µL 0.1 × Sample Buffer 2 and further processed according to manufacturer’s instructions. Jess instrument was used to analyze the samples (Protein Simple). Actin was used as the loading control (MAB8929, 1:1000) and detected with anti-mouse HRP. mCh-Trim7 was detected with rabbit polyclonal anti-mCherry (ab167453, 1:200), and EGFP-GYG1 was detected with rabbit polyclonal anti-GFP (NB600-303, 1:1000), both detected with anti-rabbit NIR secondary antibody. Protein levels were normalized to actin and to the fraction of cells transfected (determined with the incuCyte as the fraction of cells with detectable green fluorescence).

### 2.12. In Vitro Ubiquitination Experiment

In vitro ubiquitination reactions were carried out in 1× ubiquitination buffer (50 mM Tris⋅HCl, pH 7.4, 2.5 mM MgCl_2_, 0.5 mM DTT) with the addition of 2 mM ATP, 0.5 µM His-E1, 1 µM Ube2W, Ube2N/Ube2V2, 8 µg ubiquitin (Ub) and 400 ng TRIM7-RING. Reaction mixtures were incubated at 37 °C for 1–4 h, quenched by the addition of LDS sample buffer and boiling at 95 °C for 5 min. Samples were resolved by LDS-PAGE and detected by immunoblot with TRIM7 or anti-Ub-HRP (Santa Cruz, sc8017-HRP P4D1, 1:1000) antibodies.

### 2.13. Ubiquitin Discharge Assay

To prevent auto-ubiquitination, lysine 92 of Ube2N was replaced with arginine [33]. Ube2NK92R was loaded with ubiquitin by mixing 40 µM of the E2 with 1 µM Ube1, 0.37 mM Ub and 3 mM ATP in 50 mM HEPES pH 7.5, 150 mM NaCl, 20 mM MgCl_2_ and incubating the reaction at 37 °C for 30 min. The reaction was transferred to ice and used immediately. To observe E3 mediated discharge of ubiquitin, 2 µM ubiquitin-loaded E2 was mixed with 1.5 µM TRIM7-RING in 50 mM HEPES pH 7.5, 150 mM NaCl, 20 mM MgCl_2_, 50 mM L-lysine, 2.5 µM Ube2V2. Samples were taken at the time points indicated in the text and mixed immediately with LDS sample buffer at 4 °C. The samples were boiled for exactly 20 s, resolved by LDS-PAGE and observed by immunoblot using anti-Ube2N (Bio-Rad, Hercules, CA, USA, AHP974, 1:1000) and detected by near-infrared detection (Odyssey, LI-COR, Lincoln, NE, USA).

### 2.14. Western Blotting

Stable cell lines were confirmed for overexpression of TRIM7 by Western blotting. Cells were lysed in RIPA buffer supplemented with 1 × complete protease inhibitors (Roche). Lysates were separated on a precast SDS-PAGE gel (NuPAGE, 4–12%), transferred onto nitrocellulose membranes and blotted for GAPDH (Ambion, AM4300; 1:20,000) and TRIM7 (generated in house, αTrim7; 1:250)

### 2.15. Antibody Generation and Animal Handling

C57BL/6 wild-type mice were obtained from Jackson Laboratories and bred in LMB ARES facility under SPF conditions. Thirteen-week-old mice were used for the immunization experiment, which was conducted in accordance with the moderate severity limit protocol 7 and Home Office Animals (Scientific Procedures) Act (1986). All animal work was licensed under the UK Animals (Scientific Procedures) Act, 1986 and approved by the Medical Research Council Animal Welfare and Ethical Review Body. Mice (n = 12) were immunized subcutaneously (s.c) with 100 μg TRIM7-Ring-Box protein in PBS mixed with complete Freund’s adjuvant, 200 μl of emulsion per animal. Mice received 3 rounds of s.c boosting with 50 μg protein mixed with incomplete Freund’s adjuvant. Boosting was done on days 21, 42 and 79. Tail bleeds for ELISA analyses were collected on days 31, 58. Mice were sacrificed on day 93, and intracardiac blood was collected.

### 2.16. ELISA

A 96-well plate (Nunc) was coated overnight with 1 μg/mL of TRIM7-RING-BOX. The plate was blocked with 2% Marvell in PBS, 0.05% Tween 20 (MPBST) and incubated with diluted mouse sera. Bound antibodies were detected with goat anti-mouse IgG-GRP (Jackson Immunoresearch, 115-035-071).

### 2.17. Peptides

The peptides DNIKKKLDTYLQ and NLGLSMLLQ were ordered from St John’s Innovation Centre, Cambridge, UK. Peptides DNIKRKLDTYLQ, TIEALFQ, AIEALFQ, TIEALFA, TIEALAQ, TIEALRQ, TIEALVQ, TIEALFE, TIEALEQ, LQ, LLQ, Ac-LLQ, AAAAAAA, AAAAALQ were synthesized by DesignerBioscience (Cambridge, UK), dissolved in DMSO to 50–200 mM and stored at −80 °C. Peptides HDDFGLQ, LEALEFQ were synthesized by GenScript (Oxford, UK) and dissolved in DMSO to 50 mM and stored at −80 °C.

### 2.18. Isothermal Titration Calorimetry (ITC)

Titrations were performed using the MicroCal iTC200 and MicroCal Auto-iTC200 instruments (Malvern Panalytical, Malvern, UK). Binding assays were carried out in 150 mM NaCl, 50 mM Tris pH 8, 1 mM DTT and 0–1% *v*/*v* DMSO. Generally, 13–20 injections were carried out with a DP of 6 μcal/s, 750 RPM, 20–25 °C, 150–180 s spacing and 5–6 s injection times of 2.5–3 μL. Titrations were performed with protein and peptide in cell and syringe interchangeably. Generally, TRIM7 in the cell was kept at 20 μM with peptides diluted to 400 μM and up to 2 mM for dipeptides. See Appendix A for details. Control titrations of titrant into buffer were carried out where appropriate. Data were analyzed using MicroCal PEAQ-ITC analysis software, using one site model to fit the data. For low c-value experiments (such as the LQ titration), the N was fixed to 1. For experiments with MBP-T7CCPS, the buffer was 165 mM NaCl, 50 mM Tris pH 8, 1 mM DTT, 0.5% *w*/*v* glycerol and 0–0.8% *v*/*v* DMSO. For experiments with rbGYG, mGYG and mTrim7-PRYPSRY, the protein samples were dialysed against 50 mM phosphate buffer, pH 6.5 and 1 mM DTT. ITC experiments were conducted on MicroCal ITC 200 at 15 °C or an AutoITC and analysed using a standard one-state model within MicroCal instrument software.

### 2.19. Differential Scanning Fluorimetry

Thermal stabilization assays were performed using the NanoTemper Prometheus NT48 instrument (Nanotemper Technologies, München, Germany). All experiments were performed in 150 mM NaCl, 50 mM Tris pH 8, 1 mM DTT and 1% *v*/*v* DMSO unless otherwise noted. Trim7 PRYSPRY was used at 10 μM and peptides at 90 μM. Samples were heated at 2 °C/min from 15 to 95 °C. Data were analyzed using the NanoTemper Prometheus Control Software, and the first derivative of the melt curves used to define the Tm for the apo and complexed samples. Independent experiments used the same peptide and protein stocks.

### 2.20. Crystallography

TRIM7 PRYSPRY protein was dialyzed against 20 mM HEPES, 50 mM NaCl, 1 mM DTT, pH 8.0) and concentrated to 15 mg/mL; 4 mM peptide was added and the complex crystallized in 0.2 M KCSN, 15% PEG 4000, 0.1 M NaOAc, pH 5.5 for GYG1 (NH2-DNIKRKLDTYLQ-COOH, 100 mM in DMSO), for RACO-1 (NH2-NLGLSMLLQ-COOH, 50 mM in DMSO) in 1.5 M NaPO_4_H_2_ and 1 M Sodium Citrate pH 5.5. For 2BC, Trim7 PRYSPRY at 13.15 mg/mL in 150 mM NaCl, 50 mM Tris pH 8 and 1 mM DTT was mixed 100:1 with the 2BC peptide (NH2-TIEALFQ-COOH, 100 mM in DMSO) and crystallized in 1 M Sodium Potassium Tartrate, 0.1 M HEPES pH 6.8. For the NS6 peptide (NH2-LEALEFQ-COOH, 50 mM in DMSO), the protein was mixed 50:1 and co-crystallized in 1 M Na/K Phosphate pH 5. Crystals were cryoprotected with reservoir solution containing 20–30% *v*/*v* glycerol. Data for GYG1 was collected on an in-house FR-E SuperBright high-brilliance rotating anode linked to an automated crystal mounting system (ACTOR; Rigaku, Tokyo, Japan) and processed with CCP4i [34] package as follows: data were indexed in iMOSFILM, point group was determined in POINTLESS and scaled in SCALA. Molecular replacement was performed with PHASER [35] using the TRIM21 PRYSPRY model 2IWG. After the first iteration, further structures were solved using the Trim7 model from the GYG structure. The MR model was refined with COOT [36] and REF-MAC5 [37]. Data for RACO-1 was collected at the ESRF on beamline ID23 and processed as for GYG1. Data for 2BC and NS6 were collected at the DLS (Didcot, UK) on beamline i24 and i04, respectively, processed using the FASTDP [38] and Xia2dials [39] pipeline and processed as described.

### 2.21. Phylogenetic Analysis

Coding sequences for TRIM7 and GYG1 were obtained from Ensembl v. 106 [40]. Multiple sequence alignments were constructed with Muscle v. 3.8.31 with default arguments [41]. Phylogenetic trees were reconstructed using raxml v. 8.2.12 (with “-f a -m PROTCATLGX -# 100” arguments) [42].

## 3. Results

### 3.1. TRIM7 Binds a Linear Epitope of GYG1

TRIM7 is a canonical TRIM protein with the same domain architecture of RING, B Box, Coiled-coil and PRYSPRY as the antiviral TRIM proteins TRIM5 and TRIM21 (Figure 1A). To understand how TRIM7 can interact with multiple diverse substrates, we initially focused on its originally reported ligand, rabbit glycogenin-1 (rbGYG1) [25]. Previously, the PRYSPRY domain was reported to be responsible for mediating binding [43], consistent with other TRIMs. Using recombinant proteins, we probed the interaction of TRIM7-PRYSPRY (PRYSPRY) with nested truncations of the rbGYG1 protein using ITC (Figure 1B). PRYSPRY clearly interacted with the full-length rbGYG1 with an affinity of 2.3 μM. This interaction was maintained in different orthologues of TRIM7 and GYG1 (Appendix A). Glycogenin is an ancient enzyme found in early unicellular eukaryotes, but while the catalytic domain is highly conserved, the C-terminal sequence has undergone more recent divergence. Binding to TRIM7 was maintained when the catalytic domain was deleted, suggesting a possible regulatory relationship. The non-catalytic region could be subdivided further into a variable region (VR) that differs between classes and a C-terminal section that is relatively conserved in vertebrate subphylum (‘conserved in vertebrates’; CIV). Binding was maintained to a minimal construct comprising the CIV alone (KD = 5.2 ± 0.3 µM) but lost when this region was deleted (Figure 1B and Appendix A). Structural prediction of the CIV region suggested it contains two α-helices. We, therefore, synthesized peptides corresponding to these predicted helices and found that a peptide comprising the 12 carboxy-terminal (C-term) residues (DNIKKKLDTYLQ—rbGYG1322-333) was sufficient to maintain TRIM7 binding (Figure 1B). Co-crystallization of the PRYSPRY with the equivalent peptide from human glycogenin-1 (DNIKRKLDTYLQ—hsGYG1322-333) confirmed binding and the helical nature of the GYG322-333 epitope (Appendix A, Figure 1C, left). TRIM7 binding takes place at the same location within the PRYSPRY as used by TRIM21 PRYSPRY to bind its ligand, IgG Fc, further indicating that this is a specific interaction (Figure 1C, right).

### 3.2. Cellular and Viral Substrates of TRIM7 Terminate with a Glutamine

Having identified the precise epitope within GYG1 bound by TRIM7, we searched for a similar sequence in TRIM7’s other reported ligands. RACO-1 contains no similar epitope, but we noted that it also has a C-terminus ending in ‘LQ’. TRIM7 has also been reported as a restriction factor for the picornavirus CVB3 and the calicivirus MNV. The genome of picornaviruses, and many other positive-stranded RNA viruses such as SARS-CoV-2, encode their major replicase components as a single polypeptide that is subsequently processed into individual proteins by the viral 3C protease (3Cpro). Importantly, 3Cpro cleaves after glutamine (Q), meaning this is the C-terminal residue of many of the viral proteins. CVB3 contains eight proteins ending in Q, while MNV1 contains two (Figure 2A). We hypothesized that this C-terminal Q may be the motif that TRIM7 recognizes in all its reported ligands, explaining its promiscuous binding. To test this, we synthesized peptides corresponding to the C-termini of RACO-1, CVB3 2BC, which was previously identified as a possible ligand [20], and both MNV NS6 and NS3 (Figure 2B and Appendix A). Remarkably, in ITC experiments with TRIM7 PRYSPRY, all peptides bound with comparable affinity (Figure 2D, Appendix A).

### 3.3. Crystals Structures of TRIM7 Bound to Substrate Peptides Reveal a Common ΦQ Binding Mechanism

Solving the co-crystal structures of each complex (Figure 2B–D and Appendix A) revealed that the different peptides are bound in the same PRYSPRY pocket, with the penultimate two residues responsible for most of the binding interaction. In each case, there is the potential for hydrogen bonds between the terminal glutamine residue and TRIM7 residues N383, R385, G408, Q436 and S499 (Figure 2C,D). An additional hydrogen bond is also formed between the peptidyl oxygen of the penultimate peptide residue (L or F) and the peptidyl nitrogen of TRIM7 residue T384. We tested the importance of these residues in ITC binding experiments (Figure 2D). Mutants N383A, R385A, F426 and Q436A abolished binding to all tested peptides, whilst T384A bound with reduced affinity (Figure 2D and Appendix A). Mutation L423A effectively abolished binding, albeit there was some evidence of a weak binding signal in the ITC traces for the three peptide ligands. These results are consistent with the side-chains of R385 and Q436 participating in binding and only the main chain of T384. It is surprising that the bidentate salt bridge formed with R385 by the carboxylic acid of each peptide seems to play such an important role, given this interaction has no inherent residue specificity. Interestingly, an apo structure of TRIM7 was recently solved in which one of several malonate ions coordinated by the protein mimics both this carboxylic acid:R385 salt-bridge and the hydrogen bond with the main-chain of T384 (Appendix A). Investigating this further, we found that mutating the C-terminal glutamine in the 2BC peptide to alanine abolished binding (Figure 2E). This suggests that interaction between the glutamine side-chain and Q436 in TRIM7 may be necessary to position the carboxy terminus for binding to R385. As 3Cpro occasionally cleaves after glutamate, we also tested whether this residue is also accommodated by TRIM7. However, a glutamine to glutamate substitution also abolished binding. This suggests that despite binding being driven by a seemingly nonspecific interaction involving a free carboxy termini, there is exquisite sequence specificity. The penultimate residue in 3Cpro cleavage has some variability. For instance, CVB3 proteins end with ‘AQ’, ‘RQ’, ‘EQ’, ‘HQ’, ‘FQ’ and ‘VQ’. However, binding was only maintained in an ‘FQ’ peptide (Figure 2E). Previously, residue T323 was suggested as important in TRIM7 restriction of CVB3, as a T323A mutation allowed the virus to escape TRIM7 restriction [20]. However, the T323A mutation had no effect on binding, suggesting that this escape mutant may be altering viral fitness independent of TRIM7 (Figure 2E), or disrupting binding in the context of the full protein. To confirm these results with an independent method, we tested all of our 2BC peptides for TRIM7 interaction using differential scanning fluorimetry. We only observed an increase in PRYSPRY thermostability upon the addition of 2BC peptides ending in ‘FQ’ (Figure 2F).

**Figure 2 viruses-14-01610-f002:**
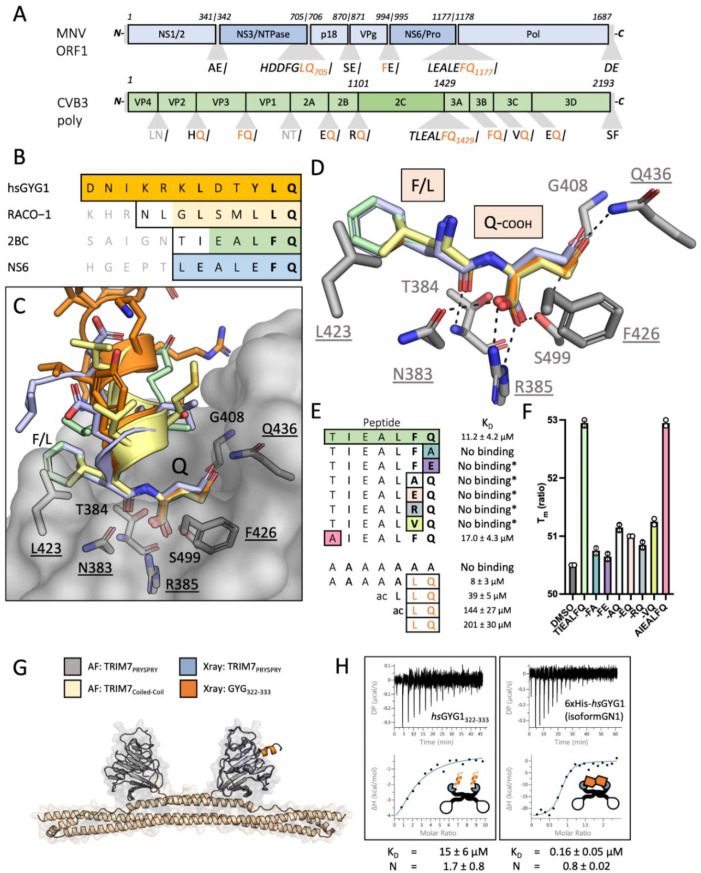
TRIM7 binds diverse substrates using a C-terminal helix-ΦQ motif. (**A**) MNV1 polyprotein (pale blue) and the relevant proteins in a darker shade (NS3 and NS6), CVB3 polyprotein in pale green and the relevant 2C protein in a darker shade. (**B**) Main peptide sequences used in the binding and structural experiments. Outlined are the sequences synthesized, whilst the shaded residues are those resolved in the crystal structures. Color coding is maintained throughout the figure. (**C**) hisTRIM7-PRYSPRY:Peptide complex structures superposed. Shows PRYSPRY as a transparent surface representation with a few key residues in stick. Peptides are color coded as described. PDB accession codes are 7OW2, 7OVX, 8A5L and 8A5M). (**D**) Detail of the PRYSPRY pocket with the recognition motif bound. Several key residues are highlighted. Dashes indicate charged or H-bonding between the peptide and the PRYSPRY residues. Underlined residues are essential for binding (see Appendix A). (**E**) Peptide substitutions based on the TIEALFQ peptide and possible polyprotein processing ends. Binding experiments with the minimal LQ motif, sequence is shown and the derived KD from ITC titrations. No binding* denotes where the peptide was only tested using nanoDSF. (**F**) Thermal denaturation data of hTRIM7-PRYSPRY derived using the Prometheus nanoDSF. Shows the Tm of the protein in the presence of peptide or DMSO. Peptides which bind stabilize the protein. The results of two independent measurements are shown. (**G**,**H**) Avidity enhancement of binding affinity with hisMBP-TRIM7-CC-PRYSPRY. (**G**) Shows the AlphaFold prediction of the TRIM7 dimer (grey and wheat) aligned with the crystal structure (pale blue and orange). (**H**) The full-length GYG1 protein (isoform GN1) shows clear 1:1 binding between the protein dimers, whilst no difference in affinity is observed when the peptide is the substrate. Representative traces and their accompanying fitted KD’s are shown.

### 3.4. Constrained C Terminal -ΦQ Motif Is Sufficient to Explain Monomeric TRIM7 Binding, but the Interaction Is More Potent When Both TRIM7 and Substrate Are Dimeric

The above data suggest that TRIM7 can interact with proteins ending in either ‘LQ’ or ‘FQ’. To determine whether these residues alone are sufficient for binding, we compared the dipeptide LQ with N-terminal acetylated LQ (ac-LQ) and ac-LLQ. These bound with increasing affinity from 200 to 39 µM. Next, we compared the binding of a simple polyA peptide with one where the last two residues were LQ (Figure 2E). While no binding could be detected for polyA, binding of 8 µM was measured for AAAAALQ—the same as that of the peptides from identified TRIM7 ligands. Taken together, these data indicate that the L/FQ alone is sufficient for TRIM7 binding, but it needs to be presented in a constrained format, as seen with increasing affinity from the dipeptide to the full protein. In the context of TRIM7 as a restriction factor, this means that CVB3 proteins VP3, 3A and 2C and MNV proteins NS3 and NS6 are possible TRIM7 ligands. More generally, our results define TRIM7 ligands as proteins or peptides containing a C-terminal helix-ΦQ motif. However, although there are many human proteins that end with ‘LQ’ or ‘FQ’, most of these are unlikely to be TRIM7 substrates as other constraints such as steric accessibility are likely to play a factor. Affinity may also be important; those measured here represent monomer binding, whereas TRIM proteins are dimers. To test how dimerization may alter target affinity, we produced MBP-TRIM7-CC-PRYSPRY (CC-PS) and compared its binding to the hGYG1322-333 peptide, and a hexahistidine-tagged GYG1 construct comprising the full-length human GN1 isoform of GYG1, which most closely matches rabbit GYG1 (Figure 2G,H and Appendix A). In the case of the peptide, the affinity was virtually unchanged (15 ± 6 µM), but the stoichiometry from the analysis (N = 1.7 ± 0.8) indicated two peptides were binding to the dimer, as expected. For the full-length protein, the affinity increased 100-fold (0.16 ± 0.05 µM) compared to the peptide or 10-fold from PRYSPRY-hsGYG1-GN1 interaction (KD = 2.1 ± 0.6), with the stoichiometry indicating binding is 1:1 between the dimers.

### 3.5. TRIM7 Contains an Active RING E3 Domain Capable of Self-Ubiquitination

TRIM7 has been shown to induce degradation of CVB3 2BC by catalyzing K48-chain ubiquitination, a signal for proteasomal recruitment [20]. This is similar to related proteins TRIM5 and TRIM21, which catalyze the ubiquitination of their substrates via their E3 RING domains, leading to substrate degradation and immune signaling via K48- and K63-linked ubiquitin chain synthesis. K48-chains are a signal for proteasomal recruitment, whereas K63-chains are immune second messengers that activate multiple signaling hubs such as TAK1 and TBK1. TRIM5 and TRIM21 first use the E2 Ube2W to monoubiquitinate their own N-termini. This monoubiquitin then acts as a primer for K63-chain extension by the hetero-dimeric E2 Ube2N/2V2. To investigate whether TRIM7 utilizes this same mechanism, we carried out a series of in vitro ubiquitination reactions using these E2s (Figure 3A,B). Incubation with Ube2W resulted in monoubiquitination of the TRIM7 RING, whilst Ube2N/2V2 did not modify the protein. However, the addition of both Ube2W and Ube2N/2V2 together resulted in the modification of TRIM7 with an extended ubiquitin chain (Figure 3C). To confirm that the TRIM7 RING is not simply the substrate for ubiquitination but is catalyzing the reaction, we performed discharge experiments and measured the kinetics of ubiquitin release from pre-charged Ube2N. The addition of TRIM7 RING substantially accelerated ubiquitin discharge from Ube2N, confirming that it is an active E3 ligase with this E2 enzyme (Figure 3D). Taken together, this shows that TRIM7 behaves similarly to TRIM5 and TRIM21 and can modify itself with K63-polyubiquitin. To demonstrate that the RING domain is active in the context of the full protein and in cells, we overexpressed different constructs of HA-TRIM7 in 293Ts (Figure 3F). Overexpression of WT and ΔPRYSPRY constructs led to laddering consistent with its ubiquitination. Laddering was abolished upon over-expression of a ΔRING construct.

### 3.6. TRIM7 Binding in Cells Leads to Codegradation with the Substrate

Next, we investigated whether TRIM7 binding of helix-ΦQ substrates leads to degradation in cells. We designed an experiment (Figure 3G) where we co-transfected cells with plasmids encoding mCherry-Trim7 (mCh-T7) constructs (WT and R385A) and EGFP-hsGYG1-GN-1 (EGFP-GYG1) constructs (WT and Q333A). mCh-T7 formed puncta, consistent with the cytoplasmic bodies formed by many TRIM proteins (Figure 3H and Appendix A). When co-expressed, EGFP-GYG1 formed puncta that co-localized with mCh-T7 (Figure 3H,I; top row). Either Q333A substitution in GYG1 or R385A substitution in TRIM7 resulted in a loss of co-localization, suggesting that GYG1 is recruited to TRIM7 puncta through direct binding. To test if TRIM7 binding causes degradation of EGFP-GYG1, we quantified cellular protein levels by fluorescence microscopy (Figure 3J) and immunoblotting (Figure 3K,L). Strikingly, EGFP-GYG1 protein levels were ~3 times higher when binding to TRIM7 was prevented with either GYG1-Q333A or TRIM7-R385 substitutions, or the TRIM7 RING domain was deleted (dR). This data supports the hypothesis that TRIM7 targets helix-ΦQ substrates for degradation by direct binding and RING-dependent degradation. Importantly, we noted that TRIM7 is co-degraded alongside GYG1. This is a hallmark of TRIM ligases, and similar co-degradation has been observed during the function of both TRIM5 and TRIM21 [44,45,46].

### 3.7. Restriction of MNV1 and CVB3 Viral Replication Is Dependent on TRIM7 Binding and Substrate Degradation but Not Its Immune Signalling Activity

As TRIM7 is reported to restrict both MNV1 and CVB3 viruses, we sought to understand if the antiviral effect was due to binding and subsequent degradation of 3Cpro-liberated viral proteins with accessible helix-ΦQ C-termini or because of immune activation. We performed a series of experiments where we expressed increasing levels of TRIM7 or known antiviral proteins TRIM5 and TRIM21, which in addition to directly restricting viruses by mediating their degradation, have also been shown to stimulate immune signaling pathways. This resulted in dose-dependent activation of both NF-κB and AP-1 pathways (Figure 4A–C). Strikingly, TRIM7 activated both pathways substantially more strongly than either TRIM5 or TRIM21 and sufficiently to induce transcription of antiviral cytokines TNF and CXCL10 (Figure 4D). Immune stimulation was dependent upon the RING domain but not the PRYSPRY, consistent with activity being induced by RING-mediated ubiquitination (Figure 4C). Consistent with this, the R385A mutant did not prevent TRIM7 from activating NF-κB (Figure 4E). Next, we generated cell lines stably expressing various TRIM7 constructs (Figure 4F and Appendix A) in HeLa cells expressing the MNV1 receptor CD300lf. These cells were then used in infection experiments with recombinant CVB3-GFP or WT MNV1 virus (Figure 4G). For CVB3, we used a target cell infection assay in which we quantified the expression of a virally-encoded GFP gene using an IncuCyte (Figure 4H). Cells expressing WT TRIM7 were able to reduce CVB3 infection by ~4-fold, but neither ΔRING, ΔPRYSPRY or R385A constructs were capable of restriction. For MNV1, we quantified the production of infectious virus from target cells by TCID50 (Figure 4I). Here the antiviral effect was even more striking, with WT TRIM7 expressing-cells reducing MNV1 infection levels by 3–4 log10. However, it is not possible to say that TRIM7 is more effective against MNV1 than CVB3, given the nature of the different assays. Activity against MNV1 was completely lost in constructs R385A or ΔPRYSPRY, where binding to the substrate was abolished. There was some loss of infection in cells expressing ΔRING, which may indicate that the binding of TRIM7 to viral proteins alone partially disrupts their function. Importantly however, because immune signaling is PRYSPRY and R385A-independent, whereas viral restriction is lost in ΔPRYSPRY and R385A cells, our data suggest that TRIM7 antiviral activity is dependent upon binding and degradation and not activating innate immunity. Many ssRNA viruses produce a polyprotein that is cleaved by a virally-encoded 3Cpro, including SARS-CoV-2. Analysis of SARS-CoV-2 suggests there are 10 viral proteins that possess an ‘LQ’ or ‘FQ’ and thus are potential substrates for TRIM7 (Appendix A). To test whether TRIM7 can restrict SARS-CoV-2, we used the same TRIM7/CD300lf HeLa cells that restrict both MNV1 and CVB3 and stably transduced them with hACE2 to make them permissive for SARS-CoV-2 infection (Appendix A). We infected these hACE2/TRIM7/CD300lf cells with the omicron variant BA.1 (EPI_ISL_7886688) at an MOI of 0.2 TCID50(Appendix A). After 18 h, we measured viral replication by qPCR using probes against genomic viral RNA encoding N protein [31,32]. Unlike with MNV1 and CVB3, we observed no TRIM7 restriction of SARS-CoV-2 replication. This is in contrast to our previously published work with SARS-CoV-2, where we observed loss of infection in the presence of neutralizing antibodies [31,47] or reduced replication upon knockout of the cellular co-factor furin [29]. However, we do not rule out that under certain conditions or in different cell types, over-expressed TRIM7 may be capable of restricting SARS-CoV-2.

## 4. Discussion

The data presented here show that TRIM7 targets diverse substrates through a highly unusual binding mechanism in which it recognizes a C-terminal helix-ΦQ motif. This mechanism explains the promiscuous binding reported for TRIM7 against unrelated proteins GYG1, RACO-1, 2C, NS3 and NS6. During preparation of our manuscript similar data were deposited in bioRxiv [48], showing that TRIM7 binds to a ΦQ motif. The helix-ΦQ motif recognized by TRIM7 is reminiscent of the N-end degron system. In the ‘N-end rule’ system [49], N-recognin proteins hydrogen bond with free α-amino groups at protein N-termini [50]. TRIM7 recognizes the other end of proteins, using an arginine residue (R385) to form a bidentate salt bridge with the carboxylic acid at C-termini. Both systems also share similarities in how substrates can be generated. In the N-end system, proteolytic cleavage of N-terminal methionine exposes amino acids at the termini that differ in their suitability as substrates for degradation. Likewise, proteolysis can reveal novel substrates for TRIM7 by exposing C-terminal glutamines. We propose that it is the action of 3C proteases that generates viral substrates for TRIM7 and results in the restriction of CVB3 and MNV1 infection. In support of our hypothesis, we show here that when C-terminal glutamine binding is abolished by mutating TRIM7 residue R385, CVB3 and MNV1 are no longer restricted.

As many positive-stranded RNA viruses utilize a 3C or 3C-like mechanism to liberate their proteins from an initial single polypeptide, there is the possibility that TRIM7 could function as a more broadly-acting restriction factor than antiretroviral ligase TRIM5. However, the ends generated by 3C proteases are not strictly conserved. For example, human norovirus proteins do not have identical C-termini as MNV1 proteins (e.g., in Norwalk virus, NS6 ends in EGETALE rather than LEALEFQ, but NS3 maintains a terminal LQ). Moreover, not all viral proteins with a helix-ΦQ motif will be accessible in the full-length form or if the protein is sequestered during viral replication in the cells. Nevertheless, it is surprising that we observed no restriction of live SARS-CoV-2, as the virus possesses 10 proteins ending in ‘LQ’ or ‘FQ’. We purposefully performed our experiments in the same cells where we had observed restriction of both MNV1 and CVB3 so we could be sure there was functional TRIM7 present. However, it is possible that restriction requires a cell line that is more physiologically relevant for SARS-CoV-2 or that expresses even higher levels of TRIM7. During preparation of this manuscript, a pre-print report was submitted that claims TRIM7 can bind to and degrade multiple SARS-CoV-2 proteins, including NSP5 and NSP8 [48]. If similar conditions can be recreated during an infection, then perhaps restriction would be observed. Of course, it may also be that binding does not happen during live virus infection, or there is insufficient degradation to impact viral replication. Alternatively, SARS-CoV-2 may antagonize TRIM7 function.

An important question raised by this study is what the principle physiological function of TRIM7 actually is. Just because TRIM7 possesses antiviral activity does not mean this is its natural cellular function, particularly as we show such activity is dependent upon a highly promiscuous binding mechanism that makes ~1% of cellular proteins potential substrates. Both CVB3 and MNV viral screens were based on TRIM7 over-expression, as were many of the other reports identifying TRIM7 ligands. Ectopic TRIM7 over-expression may have created an opportunity for binding and ubiquitination that does not exist under normal physiological conditions. The link between TRIM7s helix-ΦQ binding mechanism and 3C-protease processing, while highly attractive, may be coincidental. A decreased CVB3 viral yield was reported in mouse myoblasts and cardiomyoctes treated with TRIM7 siRNAs [20], however, this may have been an indirect effect caused by reduced cell growth and metabolism in the depleted cells. TRIM7s link to glycogenin makes this a particularly valid possibility. We strongly suggest that future studies of TRIM7 restriction phenotypes, such as with SARS-CoV-2, avoid overexpression and use a knockout approach to avoid possible artefacts. Even so, it will be important to rule out confounding factors such as altered cell growth, and ultimately viral challenge experiments in TRIM7 knockout animals may be required for a definitive answer.

We have shown here that in addition to its reported degradation activity, over-expressed TRIM7 stimulates signaling pathways, including AP-1 and NF-κB, by synthesizing K63-polyubiquitin. Dual signaling/effector function is a signature of bona fide antiviral TRIMs, TRIM5 and TRIM21. Indeed, TRIM-based overexpression screens have previously identified multiple TRIMs with antiviral restriction and NF-κB activity [19]. However, it is possible that overexpression creates artefactual conditions for signaling. Over-expression can push TRIM proteins into forming cytoplasmic bodies that would not normally assemble under resting conditions [45], and these bodies are a source of ubiquitination and downstream signal transduction.

Phylogeny and tissue expression offer a useful perspective into TRIM7 function. TRIMs have undergone a rapid expansion in mammals: there are over 70 TRIMs in humans, whereas only 20 in worms and 10 in flies [51]. Recently expanded TRIMs comprise a sub-group that evolves at a faster rate and contains a PRYSPRY domain [52]. Positive selection has been identified in many of these proteins, and while this includes TRIMs that have no known antiviral role, such a signature can be indicative of pathogen evolutionary pressure [53]. TRIM7 contains a single positively selected site in its coiled-coil, in contrast to the 21 sites found in antiretroviral TRIM5 [53]. TRIM7 is also older than antiviral TRIMs such as TRIM5 and TRIM21, which are only found in mammals. The evolutionary age of TRIM7 is a reasonable match to the appearance of the helix-ΦQ motif in glycogenin. Glycogenin arose in early eukaryotes, but the final ~30 residues did not become fixed until jawed vertebrates and the helix-ΦQ motif only in amniotes (~319 Mya; Appendix A and [54]). TRIM7 is present in birds and reptiles, suggesting it is slightly younger (~261 Mya; Appendix A and [54]). Tissue expression is another useful predictor of function, and in this respect TRIM7 and glycogenin are also a good match. TRIM7 is expressed principally in skeletal muscle and in the brain, both regions where glycogenin is also highly expressed (Appendix A). TRIM7 expression is highly tissue-specific, unlike antivirals TRIM5 and TRIM21, which are expressed almost everywhere. Moreover, TRIM7 tissue expression does not match that of the MNV1 receptor CD300lf, the CVB3 receptor CXADR or the SARS-CoV-2 receptor ACE2 (Appendix A), which would be strange if it had evolved to restrict these viruses. Finally, in contrast to Trim5 and Trim21, the Trim7 gene does not possess either an interferon-stimulated response element (ISRE) or gamma interferon-activated site (GAS), and its expression is not upregulated by interferon [55]. Taken together, neither phylogeny, tissue expression nor gene regulation are particularly supportive of TRIM7 being a broad-acting antiviral restriction factor. It is possible that none of the currently identified TRIM7 interacting proteins are the natural substrate—indeed, the real TRIM7 substrate may not be a protein at all but a metabolite. The malonate observed to bind apo TRIM7 makes the same core salt bridge as the glutamate side chain and carboxylic acid in the protein ligands. However, this moiety is not sufficient for full binding affinity, meaning that any natural metabolite would likely need to make additional interactions. Typically, an enzyme’s function is revealed with its substrate, but TRIM7s unusual binding mechanism obviates such an approach. Future work will need to focus on TRIM7 in the tissues where it is naturally found, using knockout-out approaches and whole organism models.

## Figures and Tables

**Figure 1 viruses-14-01610-f001:**
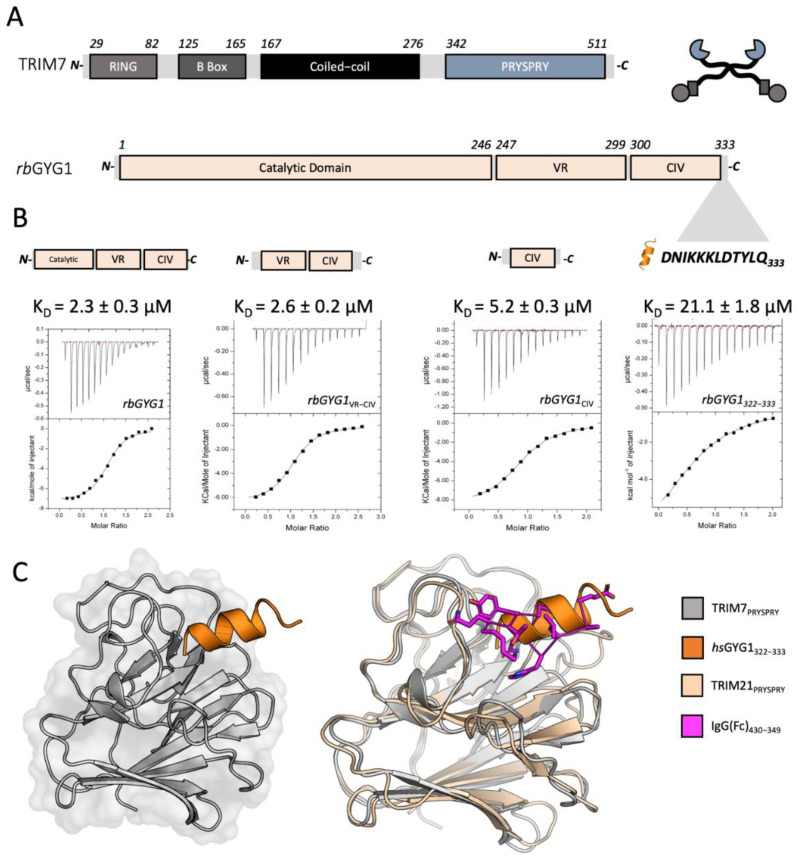
A linear epitope of GYG1 is sufficient to explain TRIM7 binding. (**A**) Sequence and domain organization of TRIM7 (grey) and Glycogenin1 (pale orange). TRIM7 is also depicted as a cartoon (circle—RING, rectangle—B-Box, line—Coiled-coil, cut-out circle—PRYSPRY) used throughout this paper. Highlighted are sequences determining TRIM7 binding. (**B**) ITC binding experiments with sequential truncations of rbGYG1 (shown above each titration) titrated into human hisTRIM7-PRYSPRY. Representative traces and their accompanying fitted KD’s are shown. (**C**) Cartoon overview of the crystal structure of hisTRIM7-PRYSPRY (grey) with the GYG1 peptide (orange) on left (PDB accession 7OVX). Comparison with TRIM21:Fc structure (2IWG, PRYSPRY wheat, Fc magenta) on the right.

**Figure 3 viruses-14-01610-f003:**
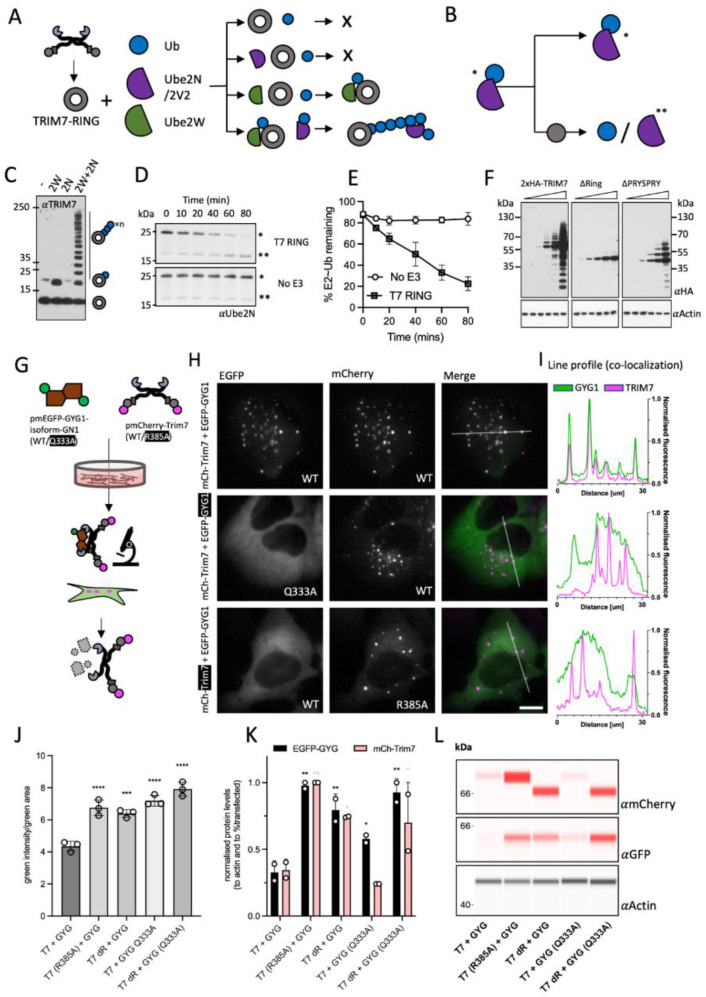
TRIM7 co-localizes with helix-ΦQ containing substrates inside cells and degrades them. (**A**–**E**) Schematic representation of in vitro ubiquitination experiments. (**A**) TRIM7-RING (grey circle) was mixed with combinations of Ubiquitin (Ub, blue circle), Ube2N/Ube2V2 (purple semi-circle) and Ube2W (green teardrop) as shown and the reactions followed by immunoblotting (**C**) with an anti-TRIM7 antibody. A representative blot from at least three independent experiments is shown. (**B**) E2 discharge experiment: Ube2N~Ub complex was incubated with TRIM7-RING, and the reaction followed over time as indicated in (**D**). Blots were probed with anti-ubiquitin antibody. A single asterisk denotes the charged Ube2N~Ub, whereas a double asterisk denotes the uncharged Ube2N. A representative blot from at least three independent experiments is shown. (**E**) Densitometry quantification of band intensities from 3D was plotted to show the kinetics of E2-Ub discharge. Error bars in all graphs depict the mean +/− SEM. Data represent three independent replicates. (**F**) Western blots from cells overexpressing indicated constructs of epitope-tagged TRIM7 (probed with anti-HA antibody). Ubiquitin-laddering is lost when the RING domain is deleted. A representative blot from two independent experiments is shown. (**G**) Schematic overview of the experiments presented in (**H**). EGFP-GYG1 is represented as a green-brown shape. TRIM7 is shown as usual but with N-terminal mCherry (magenta circle). Plasmids were co-expressed, and the fluorescence monitored using live imaging and the protein levels quantified using either the fluorescence intensity or using cell lysates. (**H**) Live-cell microscopy of U2OS cells expressing mCherry-TRIM7 and EGFP-GYG1 constructs. Left column shows the EGFP signal (green), middle column shows the mCherry signal (magenta) and the right column shows the false-colored merged image (EGFP—green; mCherry—magenta; merge—white). The scale is the same in all images, and the bar represents 10 µm. Rows represent different conditions: Top is both WT sequences. Middle has a Q333A mutation in the EGFP-GYG1 construct. Bottom has the R385A mutation in the mCh-TRIM7 con-struct. Example images are shown from at least two independent experiments. (**I**) Line profile analysis (ImageJ) of the fluorescent signal. Green trace shows the EGFP signal whilst the magenta trace shows the mCherry signal. The line used in the analysis is shown on the merged signal images. (**J**) Fluorescence-based quantification (using the IncuCyte) of EGFP-GYG1 protein levels, graph shows the total integrated intensity of the EGFP signal divided by the EGFP area from three biological replicates. ANOVA was used for statistical analysis and significant differences from T7 + GYG condition indicated (*p* < 0.0005 (***), *p* < 0.0001 (****)). (**K**,**L**) Protein quantification and blots using cell lysates and capillary-based Western Blot (Jess). Values from two biological replicates were normalized to loading control (actin) and the fraction of cells transfected (see methods). ANOVA was used for statistical analysis and significant differences from T7 + GYG condition indicated (*p* < 0.05 (*), *p* < 0.005 (**)). Black asterisks indicate significance in EGFP-GYG expression and pink asterisks in mCh-TRIM7 expression.

**Figure 4 viruses-14-01610-f004:**
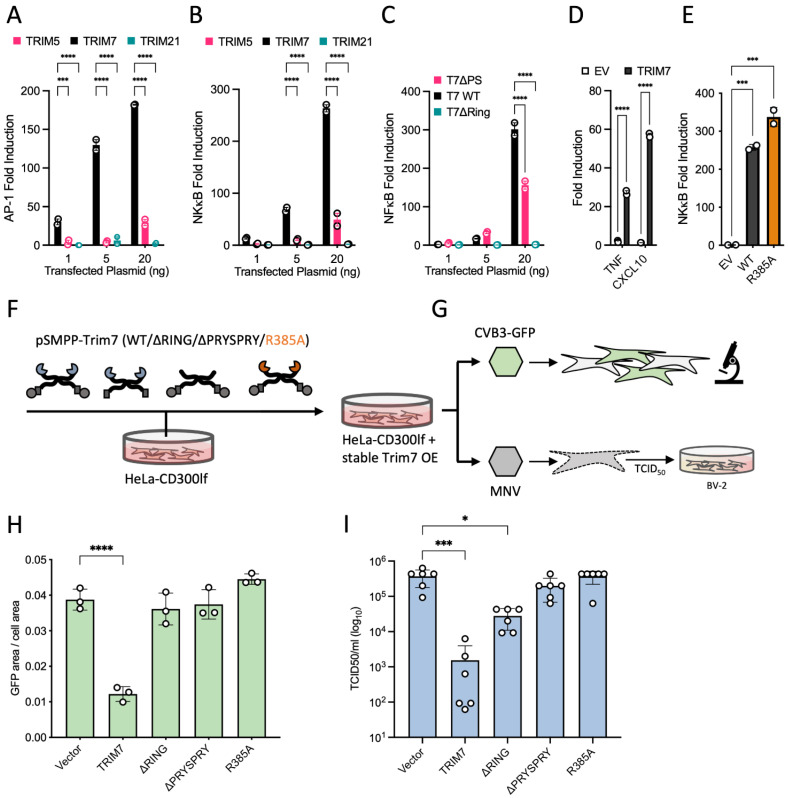
TRIM7 restriction of MNV1 and CVB3 infection is driven by helix-ΦQ binding. (**A**–**C**) Co-transfection of TRIM constructs and reporter plasmids. Overexpression of TRIM7 induces strong signaling by AP-1 and NF-κB that is dependent on the RING domain. (**D**) Overexpression of TRIM7 induces signaling, as measured by qPCR; values are normalized to actin. (**E**) Point mutation that prevents binding to targets does not impact signaling. For all signaling experiments (**A**–**E**), representative examples of at least two independent experiments are shown, with values normalized to cells transfected with empty vector. ANOVA was used for statistical analysis and significant differences indicated (*p* < 0.0005 (***), *p* < 0.0001 (****)). (**F**,**G**) Schematic overview of the experiments shown in H and I. Lentivirus generation and stable transfection of HeLa-CD300lf cells (**F**), with the expression confirmed by Western blotting (Appendix A). Cells were infected with recombinant CVB3 virus-producing EGFP in host cells or MNV1 (**G**). (**H**) Quantification of CVB3 infection by measuring the fraction of EGFP-expressing cells. Data shown are from three independent experiments. (**I**) Quantification of MNV1 replication by TCID50 following infection of TRIM7-expressing cells. Virions from lysed cells were titrated onto susceptible BV-2 cells. Data shown are a representative result of three independent experiments. Error bars show the standard deviation. Non-parametric ANOVA was used for statistical analysis and significant differences indicated (*p* < 0.05 (*), *p* < 0.0005 (***), *p* < 0.0001 (****)).

## Data Availability

Data used in the study is contained within the article and the Appendix A. Crystallographic data were deposited to the PDB. Any additional information available on request.

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
