# Peer review of "TRIM7 Restricts Coxsackievirus and Norovirus Infection by Detecting the C-Terminal Glutamine Generated by 3C Protease Processing"

_viruses, 2022, doi:10.3390/v14081610_

Round 1

Reviewer 1 Report

In this study, Luptak et al describe a variety of biochemical/structural experiments that reveal the unusual mechanism through which the TRIM7 PRYSPRY recognizes potential ligands. The execution is high quality, the descriptions are clear and the subject will be of interest to the readers of viruses (as it was to this reviewer). In addition, key findings are validated by a competing preprint from March this year, solidifying many of the conclusions. Because the mechanistic data would not be out of place in a broad-interest journal, I have made a number of suggestions that might improve the manuscript but I don’t believe any of them are essential as the paper is suitable for publication in Viruses in its current form.

Major

While this is clearly a matter for the Editor(s)/authors, it would seem that the title could be revised. I agree with the authors that there is doubt regarding whether TRIM7 is a restriction factor at all (something the authors have presented and discussed fairly) and the magnitude of coxackievirus restriction described in this study is underwhelming. This comment is not intended to diminish the value of the study but it seems that re-wording the title to reflect some of the ambiguity and/or focus on the mechanistic insight could be of value and might future-proof the study.

In relation to the comment above, the mechanistic component of this study is much more developed than the biological relevance. One point that could be strengthened relates to the comment that “These features, together with TRIM7’s restricted tissue expression and lack of immune regulation suggest that viral restriction may not be its physiological function”. The authors present a reasonable argument that the antiviral activity of TRIM7 might be a “red herring”. A loss of function/KO experiment is probably beyond the scope of the revisions, but it would be nice to solidify the argument about the tissue distribution of TRIM7 expression and immune regulation. This could preferably be done with a presentation of meta-analysed data, for example by comparing TRIM7 expression to other TRIMs (like TRIM5, TRIM25 and TRIM21) with regards to tissue distribution (like the GTEx analysis in https://doi.org/10.1016/j.cell.2020.10.030) and IFN responsivity. Alternatively, more discussion and citations would address this.

The global interest in SARS-CoV-2 means that a potential inhibitory role for TRIM7 will be of interest (particularly if the final version of the competing manuscript presents data suggesting SARS-CoV-2 inhibition). It would be nice to see a positive control inhibitor (IFN or remdesivir etc) in S6C or to include an additional TCID50-like assay for SARS-CoV-2 in the presence of TRIM7 (although this might require a different variant or new cells and be time consuming). Alternatively, the argument that potential SARS-CoV-2 inhibition requires more examination could be developed further and the language softened to be less definitive in statements like “However, we show that despite possessing multiple proteins ending in glutamine, SARS-CoV-2 is not restricted by TRIM7”.

Minor/typos

Many symbols have been replaced with a spiral in the version I was reading, including the motif, the temperatures, micrograms etc.

Useful to indicate the TRIM on panels S1 D and C

Define ac-LQ at first use in the results.

It would be nice to see a few more cells or some lower power images in 3H.

The order of the keys in 4A and 4B are different, one key or the 4B one twice would suffice.

The author contributions could be more detailed (funding acquisition, supervision, writing reviewing and editing etc.).

Author Response

We are grateful to Reviewer 1 for their comments and have provided a point-by-point reply below:

Reviewer 1

In this study, Luptak et al describe a variety of biochemical/structural experiments that reveal the unusual mechanism through which the TRIM7 PRYSPRY recognizes potential ligands. The execution is high quality, the descriptions are clear and the subject will be of interest to the readers of viruses (as it was to this reviewer). In addition, key findings are validated by a competing preprint from March this year, solidifying many of the conclusions. Because the mechanistic data would not be out of place in a broad-interest journal, I have made a number of suggestions that might improve the manuscript but I don’t believe any of them are essential as the paper is suitable for publication in Viruses in its current form.

Major

While this is clearly a matter for the Editor(s)/authors, it would seem that the title could be revised. I agree with the authors that there is doubt regarding whether TRIM7 is a restriction factor at all (something the authors have presented and discussed fairly) and the magnitude of coxackievirus restriction described in this study is underwhelming. This comment is not intended to diminish the value of the study but it seems that re-wording the title to reflect some of the ambiguity and/or focus on the mechanistic insight could be of value and might future-proof the study.

We have considered this point but decided to leave the title as is to reflect the fact that whilst the mechanism is definitive the status of TRIM7 as a bona fide restriction factor is not.

In relation to the comment above, the mechanistic component of this study is much more developed than the biological relevance. One point that could be strengthened relates to the comment that “These features, together with TRIM7’s restricted tissue expression and lack of immune regulation suggest that viral restriction may not be its physiological function”. The authors present a reasonable argument that the antiviral activity of TRIM7 might be a “red herring”. A loss of function/KO experiment is probably beyond the scope of the revisions, but it would be nice to solidify the argument about the tissue distribution of TRIM7 expression and immune regulation. This could preferably be done with a presentation of meta-analysed data, for example by comparing TRIM7 expression to other TRIMs (like TRIM5, TRIM25 and TRIM21) with regards to tissue distribution (like the GTEx analysis in https://doi.org/10.1016/j.cell.2020.10.030) and IFN responsivity. Alternatively, more discussion and citations would address this.

We are very grateful to the reviewer for these suggestions. We have incorporated a GTEx analysis as a supplementary figure in the revised manuscript.

The global interest in SARS-CoV-2 means that a potential inhibitory role for TRIM7 will be of interest (particularly if the final version of the competing manuscript presents data suggesting SARS-CoV-2 inhibition). It would be nice to see a positive control inhibitor (IFN or remdesivir etc) in S6C or to include an additional TCID50-like assay for SARS-CoV-2 in the presence of TRIM7 (although this might require a different variant or new cells and be time consuming). Alternatively, the argument that potential SARS-CoV-2 inhibition requires more examination could be developed further and the language softened to be less definitive in statements like “However, we show that despite possessing multiple proteins ending in glutamine, SARS-CoV-2 is not restricted by TRIM7”.

Given the time constraints and in lieu of an IFN control, we have referenced our other published studies where we have shown reduced SARS-CoV-2 infection using the same assay. We have also softened the language as suggested; it’s quite likely that someone will report a phenotype.

Minor/typos

Many symbols have been replaced with a spiral in the version I was reading, including the motif, the temperatures, micrograms etc.

We apologise for this problem and hope this is now fixed in the formatting process.

Useful to indicate the TRIM on panels S1 D and C

This has been corrected.

Define ac-LQ at first use in the results.

This has now been defined.

It would be nice to see a few more cells or some lower power images in 3H.

We have provided additional examples in a new supplementary figure.

The order of the keys in 4A and 4B are different, one key or the 4B one twice would suffice.

This has been corrected.

The author contributions could be more detailed (funding acquisition, supervision, writing reviewing and editing etc.).

This detail has now been added.

Reviewer 2 Report

In the manuscript, “TRIM7 restricts Coxsackievirus and norovirus infection by detecting the C-terminal glutamine generated by 3C protease processing”, Luptak et al present data indicating that TRIM7 targets a specific  C terminal motif that is present in numerous cellular proteins and in viral proteins generated by specific viral proteases, such as coxsackievirus and norovirus 3C. The authors use several biochemical and structural approaches to demonstrate that a C terminal glutamine preceded by a hydrophobic residue serves as the ligand capture site for the TRIM7 PRY/SPRY domain. Their findings confirm a recent study posted on Biorxiv showing that TRIM7 recognizes a similar C terminal degron motif.

The experiments are well executed, and the data trends generally support principal conclusions made from the biochemistry. However, several experiments do not appear to have replicates or statistics, which should be addressed. Additionally, the authors suggest multiple times throughout the manuscript that they don’t think TRIM7 functions in viral restriction and that previous work is likely an artefact of overexpression. The authors do not have any data themselves to support these speculations. Additionally, it appears the authors only select certain aspects of prior literature, and perhaps ignore others, in forming their arguments.

Major comments: 

1. Replicates and statistics. Many experiments appear to have been performed only twice and it is not clear whether the data represent technical or biological replicates. 

a. How many times were the ITC experiments in Fig 1B Figure S1, S3 performed? Replicates should be indicated (or performed if there are none). 

b. Fig 2F. Only duplicates are shown. Unclear if technical or biological reps.

c. Fig 2H and S3. Were replicates done on this? Should Kd’s should be determined from average of 3 replicates? Or is it customary to report Kd from only one experiment?

d. Fig 3 westerns. Unclear how many times these were performed.

e. Fig 3 H, I. Colocalization is shown for only one or two cells, from what seems to be one experiment. Replicates and quantification of colocalization would seem appropriate.

f. Fig 3J. Statistics? Figure 3K, only two data points shown.

g. Fig 4A-E. Only two data points and no statistics.

2. The authors might consider re-examining prior TRIM7 literature more carefully and tempering their speculations and conjectures on lines 21-22, 87-90, 650-688.

a. The authors point out the very valid concern of potential overexpression artefacts, as TRIM7 was previously identified in two independent studies that used overexpression screens as the initial discovery tool. However, the authors do not acknowledge that Fan et al (PMID: 34062120) showed that silencing of endogenous TRIM7 in human hepatoma cells, mouse myoblasts, and mouse primary cardiomyocytes resulted in increased viral replication. These cells are all potentially relevant cell types for coxsackievirus and other enteroviruses. As gene silencing is the gold standard to prove that a protein can function as a restriction factor at endogenous levels, it appears the authors’ criticism of overexpression does not take everything into account.

b. The authors discuss the evolution of TRIM7 but do not mention that a previous study found TRIM7 to exhibit modest signatures of rapid evolution (PMID: 24158625 ). As this is a characteristic of many antiviral restriction factors engaged in arms races with viruses, it seems premature to disregard the possibility that TRIM7 has bona fide antiviral activity. 

c. The authors briefly mention but do not expand upon the role of TRIM7 as a host factor that ubiquitinates the Zika virus envelope protein, thus promoting infection (PMID: 3264182). This was shown by both overexpression and knockout approaches in cells and in Trim7 KO mice. Trim7 was indeed shown to be a determinant of Zika virus tropism in mice. How does this fit with the authors’ discussions downplaying a role for TRIM7 in virus-host interactions?

Summarily, the authors might consider caution when using the word “artefact”, especially since they themselves have collected no data in a physiologically relevant model to indicate that the reported activity of TRIM7 as an antiviral protein is artefactual. 

Author Response

We are grateful to Reviewer 2 for their comments and have provided a point-by-point reply below:

Reviewer 2

In the manuscript, “TRIM7 restricts Coxsackievirus and norovirus infection by detecting the C-terminal glutamine generated by 3C protease processing”, Luptak et al present data indicating that TRIM7 targets a specific  C terminal motif that is present in numerous cellular proteins and in viral proteins generated by specific viral proteases, such as coxsackievirus and norovirus 3C. The authors use several biochemical and structural approaches to demonstrate that a C terminal glutamine preceded by a hydrophobic residue serves as the ligand capture site for the TRIM7 PRY/SPRY domain. Their findings confirm a recent study posted on Biorxiv showing that TRIM7 recognizes a similar C terminal degron motif.

The experiments are well executed, and the data trends generally support principal conclusions made from the biochemistry. However, several experiments do not appear to have replicates or statistics, which should be addressed. Additionally, the authors suggest multiple times throughout the manuscript that they don’t think TRIM7 functions in viral restriction and that previous work is likely an artefact of overexpression. The authors do not have any data themselves to support these speculations. Additionally, it appears the authors only select certain aspects of prior literature, and perhaps ignore others, in forming their arguments.

Major comments: 

  1. Replicates and statistics. Many experiments appear to have been performed only twice and it is not clear whether the data represent technical or biological replicates. 

  1. How many times were the ITC experiments in Fig 1B Figure S1, S3 performed? Replicates should be indicated (or performed if there are none). 

In all cases, ITC experiments were performed at least twice. However, the standard in the field is to quote an affinity from a given titration within an experiment and sometimes to show primary data in the form of binding isotherms, which is what we have elected to do here. In our revised manuscript, we have clarified in the legends that representative traces are shown together with the fitted KD.

  1. Fig 2F. Only duplicates are shown. Unclear if technical or biological reps.

These are independent measurements but undertaken with the same stock of peptide and proteins. This is usual for such biophysical experiments.  In our revised manuscript, we have clarified in the legends that the duplicates are independent measurements and stated in the methods that this was using the same peptide and protein stocks.

  1. Fig 2H and S3. Were replicates done on this? Should Kd’s should be determined from average of 3 replicates? Or is it customary to report Kd from only one experiment?

As mentioned above, typically representative traces are shown where it is customary to report the Kd from a given titration and quote the fitting error.

  1. Fig 3 westerns. Unclear how many times these were performed.

We have added a note to indicate the number of independent experiments in the figure legend.

  1. Fig 3 H, I. Colocalization is shown for only one or two cells, from what seems to be one experiment. Replicates and quantification of colocalization would seem appropriate.

Two independent experiments were performed, together with additional repeats of particular conditions. The purpose of the line profile is not to provide a quantification (indeed we have not quoted any figure for this) but to help the reader interpret the microscopy images. This is needed because the co-localization is so complete the puncta are almost exactly on top of each other. However, we have provided additional examples of colocalization in additional cells as a new supplementary figure in the revised manuscript.

  1. Fig 3J. Statistics? Figure 3K, only two data points shown.

We have added statistics for both data sets.

  1. Fig 4A-E. Only two data points and no statistics.

We have performed statistical analysis of the data in these panels and included them.

  1. The authors might consider re-examining prior TRIM7 literature more carefully and tempering their speculations and conjectures on lines 21-22, 87-90, 650-688.

  1. The authors point out the very valid concern of potential overexpression artefacts, as TRIM7 was previously identified in two independent studies that used overexpression screens as the initial discovery tool. However, the authors do not acknowledge that Fan et al (PMID: 34062120) showed that silencing of endogenous TRIM7 in human hepatoma cells, mouse myoblasts, and mouse primary cardiomyocytes resulted in increased viral replication. These cells are all potentially relevant cell types for coxsackievirus and other enteroviruses. As gene silencing is the gold standard to prove that a protein can function as a restriction factor at endogenous levels, it appears the authors’ criticism of overexpression does not take everything into account.

These results are surprising to us, particularly the HepG2 data as TRIM7 isn't expressed in these cells. This suggests to us that the RNAi strategy used by Fan et al either had an off-target effect or led to altered cell growth, slowing metabolism and viral replication. We suggest that a knockout approach and use of mouse models will be needed for a definitive answer. However, we have amended the discussion to reference the RNAi depletion data of Fan et al.

  1. The authors discuss the evolution of TRIM7 but do not mention that a previous study found TRIM7 to exhibit modest signatures of rapid evolution (PMID: 24158625 ). As this is a characteristic of many antiviral restriction factors engaged in arms races with viruses, it seems premature to disregard the possibility that TRIM7 has bona fide antiviral activity. 

We thank the reviewer for suggesting this reference and have now included it, along with new text on positive selection, into the revised manuscript.

  1. The authors briefly mention but do not expand upon the role of TRIM7 as a host factor that ubiquitinates the Zika virus envelope protein, thus promoting infection (PMID: 3264182). This was shown by both overexpression and knockout approaches in cells and in Trim7 KO mice. Trim7 was indeed shown to be a determinant of Zika virus tropism in mice. How does this fit with the authors’ discussions downplaying a role for TRIM7 in virus-host interactions?

Our present manuscript explores the possibility that TRIM7 is a restriction factor. We have referenced the work on Zika but there is nothing in our data that specifically supports or refutes that study and so we’ve avoided speculating. However, we agree with the reviewer that this is an interesting and surprising finding and worthy of future investigation.

Summarily, the authors might consider caution when using the word “artefact”, especially since they themselves have collected no data in a physiologically relevant model to indicate that the reported activity of TRIM7 as an antiviral protein is artefactual. 

We appreciate this point and have been careful to use this phrase in the context of overexpression.

Reviewer 3 Report

In this manuscript, Luptak et al. analyze the substrate recognition mechanism of the E3 ligase TRIM7 in order to elcuidate the mechanism underlying the diverse roles reported for this protein, which include restriction of coxsackie- and noroviruses. Applying a combination of structural and biochemical analyses, the authors very convincingly demonstrate that the protein recognizes its substrates, including selected viral proteins, via an unusual binding motif, a C-terminal helix terminating in F/LQ. Using mutated variants of TRIM7 or substrates, they further show intracellular co-localization of TRIM7 and a prototype substrate and reduced intracellular levels of this substrate in the presence of TRIM7 dependent on the interaction motifs. They also report restriction of coxsackievirus and a norovirus in tissue culture by TRIM7, dependent on residues mediating substrate binding. In my view, this study contributes important information and also raises interesting questions that will stimulate further work. I therefore consider it clearly of interest to the readers of ‚Viruses‘.

I find the manuscript well written and the figures clear. I only have some  remarks.

My main point refers to the experiment using Sars-CoV-2. The results are used to make a specific point, namely that TRIM7 is not a universal restriction factor of +strand RNA viruses carrying processed proteins terminated by F/LQ. Data in supplementary figure S6 show that no restriction of Sars-CoV-2 was observed in a TRIM7 and ACE2 expressing Hela derived cell line under the conditions used. However, the data presented are limited (only one experiment in a rather non-physiological cell line under only one condition, no control with a restricted virus in the same cell line shown). In the discussion it is also mentioned that others (pre-publication manuscript cited as ref.21.?) have observed degradation of Sars-CoV-2 proteins carrying the recognition motif mediated by TRIM7. Resolving the issue whether there may be a broader antiviral activity, or – as the authors importantly point out in the discussion  – whether antiviral restriction is even a relevant function of TRIM7 in vivo, by more detailed analyses is of great interest, but is clearly beyond the scope of this manuscript. I would suggest to retain the data in the supplement, but point out more clearly the limited nature of the experiment shown, emphasizing the point that this merits further investigation. Please also add reference(s) in line 645-6 indicating which reports are referred to here, maybe briefly mentioning the approaches used in these studies.

Line 403: Please clarify how 2BC was „identified as a possible ligand“.

Lines 438ff:  I found this part difficult to follow, since the rationale for comparing wt and GN1 isoform has not been introduced. Please clarify.

Figure S3 and the respective text in the results section: from the data shown it is not obvious to me why a distinction is made between L423A as weakly interacting vs. N383A, F426A and Q436A as not binding.

Figure 4: the assays used for CVB3 or MNV measure different aspects (infection of the target cell vs. production of infectious virus from target cells), using different types of readouts. This should be briefly acknowledged in the results text, and I would avoid a semi-quantitative comparison (line 582ff).

The level of detail of the extensive materials and methods section is a bit heterogeneous; some parts are very clear and complete, others less so. It apears to me that some denominations of cell lines or plasmids do not match. Re-editing of this section (and possibly a table listing the plasmids generated or used, incuding refs) may improve clarity.

For Figures comprising bar graphs: please specify what the individual data points represent (independent experiments, technical replicates etc).

I an few instances (Fig. 2E, 4D), the Figure numbers used in the text do not seem to match the respective panel in the figure. Please check.

Pdf conversion: At least in my download, many special characters (greek letters, °C etc.) have been randomly replaced by strange symbols throughout the manuscript. Also, the supplementary figure legends are represented in a format that is very difficult to read.

Author Response

We are grateful to Reviewer 3 for their comments and have provided a point-by-point reply below:

Reviewer 3

In this manuscript, Luptak et al. analyze the substrate recognition mechanism of the E3 ligase TRIM7 in order to elcuidate the mechanism underlying the diverse roles reported for this protein, which include restriction of coxsackie- and noroviruses. Applying a combination of structural and biochemical analyses, the authors very convincingly demonstrate that the protein recognizes its substrates, including selected viral proteins, via an unusual binding motif, a C-terminal helix terminating in F/LQ. Using mutated variants of TRIM7 or substrates, they further show intracellular co-localization of TRIM7 and a prototype substrate and reduced intracellular levels of this substrate in the presence of TRIM7 dependent on the interaction motifs. They also report restriction of coxsackievirus and a norovirus in tissue culture by TRIM7, dependent on residues mediating substrate binding. In my view, this study contributes important information and also raises interesting questions that will stimulate further work. I therefore consider it clearly of interest to the readers of ‚Viruses‘.

I find the manuscript well written and the figures clear. I only have some  remarks.

My main point refers to the experiment using Sars-CoV-2. The results are used to make a specific point, namely that TRIM7 is not a universal restriction factor of +strand RNA viruses carrying processed proteins terminated by F/LQ. Data in supplementary figure S6 show that no restriction of Sars-CoV-2 was observed in a TRIM7 and ACE2 expressing Hela derived cell line under the conditions used. However, the data presented are limited (only one experiment in a rather non-physiological cell line under only one condition, no control with a restricted virus in the same cell line shown). In the discussion it is also mentioned that others (pre-publication manuscript cited as ref.21.?) have observed degradation of Sars-CoV-2 proteins carrying the recognition motif mediated by TRIM7. Resolving the issue whether there may be a broader antiviral activity, or – as the authors importantly point out in the discussion  – whether antiviral restriction is even a relevant function of TRIM7 in vivo, by more detailed analyses is of great interest, but is clearly beyond the scope of this manuscript. I would suggest to retain the data in the supplement, but point out more clearly the limited nature of the experiment shown, emphasizing the point that this merits further investigation. Please also add reference(s) in line 645-6 indicating which reports are referred to here, maybe briefly mentioning the approaches used in these studies.

We have added additional sentences to both the results and discussion emphasising that it may be possible to observe a SARS-CoV-2 phenotype in different cells or upon expression of even higher levels of TRIM7. However, we have cautioned against basing any future phenotype on overexpression and the importance of a knockout approach. We have also added a reference to the pre-print claiming TRIM7 degradation of several SARS-CoV-2 proteins.

Line 403: Please clarify how 2BC was „identified as a possible ligand“.

Apologies there was a reference missing, this has now been added.

Lines 438ff:  I found this part difficult to follow, since the rationale for comparing wt and GN1 isoform has not been introduced. Please clarify.

We have clarified that we used the GN1 isoform of hGYG1 as this is the one most closely matching rabbit GYG1, the originally identified ligand for TRIM7.

Figure S3 and the respective text in the results section: from the data shown it is not obvious to me why a distinction is made between L423A as weakly interacting vs. N383A, F426A and Q436A as not binding.

There is some evidence of a binding signal in the ITC traces for L423, although too weak and too low affinity to fit. We have now indicated this in the text.

Figure 4: the assays used for CVB3 or MNV measure different aspects (infection of the target cell vs. production of infectious virus from target cells), using different types of readouts. This should be briefly acknowledged in the results text, and I would avoid a semi-quantitative comparison (line 582ff).

We have clarified the different assays and inserted a sentence explicitly stating that it cannot be claimed that TRIM7 is more effective against MNV1 because of these differences.

The level of detail of the extensive materials and methods section is a bit heterogeneous; some parts are very clear and complete, others less so. It apears to me that some denominations of cell lines or plasmids do not match. Re-editing of this section (and possibly a table listing the plasmids generated or used, incuding refs) may improve clarity.

We have edited this section to improve consistency.

For Figures comprising bar graphs: please specify what the individual data points represent (independent experiments, technical replicates etc).

We have clarified this in the figure legends.

I an few instances (Fig. 2E, 4D), the Figure numbers used in the text do not seem to match the respective panel in the figure. Please check.

We have corrected the error with 4D, although we couldn’t find the mismatch for 2E.

Pdf conversion: At least in my download, many special characters (greek letters, °C etc.) have been randomly replaced by strange symbols throughout the manuscript. Also, the supplementary figure legends are represented in a format that is very difficult to read.

Our apologies for this and we hope this formatting error has now been fixed.

Round 2

Reviewer 2 Report

The authors have adequately addressed this reviewers' comments.